# Near-Infrared Spectroscopy and Mode Cloning (NIR-MC) for In-Situ Analysis of Crude Protein in Bamboo

Qingyu Sheng [1], Mariana Santos-Rivera [1], Xiaoguang Ouyang [1], Andrew J. Kouba [2] and Carrie K. Vance [1,*]

[1] Department of Biochemistry, Molecular Biology, Entomology, and Plant Pathology, Mississippi State University, Starkville, MS 39762, USA; qs133@msstate.edu (Q.S.); jms2033@msstate.edu (M.S.-R.); xouyang@gmail.com (X.O.)

[2] Department of Wildlife, Fisheries & Aquaculture, Mississippi State University, Starkville, MS 39762, USA; a.kouba@msstate.edu

[*] Correspondence: ckv7@msstate.edu

**Abstract:** This study develops Near-Infrared Spectroscopy (NIRS) and Mode-Cloning (MC) for the rapid assessment of the nutritional quality of bamboo leaves, the primary diet of giant pandas (*Ailuropoda melanoleuca*) and red pandas (*Ailurus fulgens*). To test the NIR-MC approach, we evaluated three species of bamboo (*Phyllostachys bissetii, Phyllostachys rubromarginata, Phyllostachys aureosulcata*). Mode-Cloning incorporated a Slope and Bias Correction (SBC) transform to crude protein prediction models built with NIR spectra taken from Fine–Ground leaves (master mode). The modified models were then applied to spectra from leaves in the satellite minimal processing modes (Course–Ground, Dry–Whole, and Fresh–Whole). The NIR-MC using the SBC yielded a residual prediction deviation (RPD) = 2.73 and 1.84 for Course–Ground and Dry–Whole sample modes, respectively, indicating a good quantitative prediction of crude protein for minimally processed samples that could be easily acquired under field conditions using a portable drier and grinder. The NIR-MC approach also improved the model of crude protein for spectra collected from Fresh–Whole bamboo leaves in the field. Thus, NIR-MC has the potential to provide a real-time prediction of the macronutrient distribution in bamboo in situ, which affects the foraging behavior and dispersion of giant and red pandas in their natural habitats.

**Keywords:** bamboo leaves; calibration transfer; forage grasses; giant panda; nutrition; red panda; sampling

## 1. Introduction

Forage grasses and legumes supply energy and nutrients to various ruminants and non-ruminants by providing the protein, vitamins, and minerals necessary for growth and metabolism [1]. Near-infrared spectroscopy (NIRS) has been used widely in assessing the forage quality of livestock, and in recent years, an increase in studies of forage nutrition for wild species [2–4]. Importantly, the determination of the nutritional value of forage grasses can be used to understand the nutritional drivers of wild-animal movement in situ [5]. For such studies, miniaturized NIR spectrometers designed for in-field spectral collection are more practical for real-time assessment than bench-top systems [6–8].

NIRS is a cost-effective and rapid technique for measuring the nutritional quality of many forage grasses, including bamboo [9,10]. The quantitative nutrient composition of forages can be obtained using NIRS combined with mathematical transformations to generate calibration models for nutrient prediction [11]. However, sample processing emphasizes different contributions to the spectral profile of the sample, which can then affect the model performance and prediction results. In standard practice, NIR spectra are collected from forage samples that are entirely dried at temperatures from 55 to 70 °C because samples with high moisture generate a strong water absorption signal that overlaps and obscures other spectral features, which causes peak shifts and results in non-linear

responses [1,12,13]. Several studies in soil and forage analysis have found that NIR results from wet samples have higher standard errors of calibration compared with dried samples and hence have lower prediction accuracy [13,14]. Moreover, the size, shape, and uniformity of particles in a sample can change the surface light scattering and affect the slope of the NIR spectra [15], as well as cause absorption flattening [16,17]. Under laboratory conditions, isotropic and uniform sized particles can be obtained by grinding in a cyclone mill to minimize scattering noise in the spectrum. The grinding process also increases homogenization and provides the highest accuracy in prediction equations from spectral data [18].

Although NIR spectra and model prediction of nutrient quantities from dried, highly ground samples can be more accurate, the sample processing steps require time, labor, and unwieldy equipment that is unsuitable for in situ applications. Field studies require a simplified procedure for minimal or unprocessed fresh samples. The calibration transfer of a master prediction model generated for fine-ground samples may be applicable to non-ground or even wet samples [19]. However, in some cases, the model cannot be applied directly to new datasets because the gaps between the existing and new types of samples cause variation between the spectra, leading to biased predictions. For example, different processing methods of the same sample can cause a change in the physical properties of the sample, such as particle size or surface texture [20]. Here, we propose an NIR-Mode Cloning (NIR-MC) approach that consists of calibration transfer based on mathematical and statistical techniques to correct for differences caused by various processing modes [21]. One method commonly used for correcting predictions is the simple univariate Slope and Bias Correction (SBC), in which the prediction model is developed based on the Master mode and can then be modified to correct predicted values in a satellite mode [22,23]. The SBC model transfer was applied to the leaves of bamboo, which is a wild forage grass for numerous species. The predictive performance of the SBC model transfer has been shown to be similar to that of other spectral correction methods such as direct standardization (DS), piecewise direct standardization (PDS), and external parameter orthogonalization (EPO) corrections [24–26].

Bamboo is the primary food source for the giant panda (*Ailuropoda melanoleuca*) and red panda (*Ailurus fulgens*), representing 99% and 91.4–99.1% of their diet, respectively [27–29]. Bamboos are distributed throughout the panda habitat and are cold-tolerant and ever-green [30,31], serving as the primary food source throughout the year, especially in the winter. The amount of protein and fat that either panda species can acquire from a diet of bamboo is limited because bamboo is a low-nutrition/energy food source with crude protein composing only 7.0–21.6% of total dry matter and crude fat only 1.6–4.2%, with the remaining content composed of complex carbohydrates and digestible sugars [9,32]. Giant pandas and red pandas have a relatively short gastrointestinal tract (GIT), typical of carnivores, and the low abundance of cellulases and endo-hemicellulases in the carnivore digestive system make it difficult to digest cellulose into usable nutrients or to extract the protein and fat constituents [31–33]. The digestive efficiency of bamboo leaves is less than 30% in giant pandas and red pandas [34,35], resulting in the need to consume large quantities of bamboo each day.

The variation in nutrient content across different bamboo species and habitats can affect giant and red panda diet quality and dietary selection. For example, studies have shown that captive giant pandas prefer culm over leaves in spring due to the seasonal changes in carbohydrate distribution within different parts of the bamboo plant [36]. Moreover, giant pandas shift between consuming bamboo leaves and shoots of varying bamboo species throughout the year, as the concentrations of primary energy constituents and critical elements such as calcium, phosphorus, and nitrogen shift within the plant structure, potentially influencing annual patterns of migration [37]. Red pandas also prefer bamboo leaves in the summer and fall, as leaves contain the highest levels of crude protein and fat during these seasons [32]. A better understanding of the bamboo foodscape can assist wild-population management through habitat conservation and the

establishment of protected areas to account for the seasonal foraging behavior, migration, and population distribution changes. The direct assessment of forage quality in situ for giant and red panda population management is possible by using portable NIRS and sampling processing devices. To test the NIR-MC approach, we evaluated three species of bamboo from the genus Phyllostachys that are endemic to China and that serve as the primary forage of both panda species. Once the calibration models for predicting nutritional parameters were built using spectra from highly processed samples, the NIR-MC method was applied to fit spectra from minimally processed bamboo leaves with calibration transfer and mathematical correction. Building predictive models using SBC transforms that adjust for sample processing modes and are transferable to spectra collected from bamboo leaves in situ willenable real-time assessment of the bamboo nutritional landscape of panda habitat.

## 2. Materials and Methods

### 2.1. Bamboo Sampling and NIR Spectra Collection

The leaves from three bamboo species, *Phyllostachys bissetii* (Bissetii), *Phyllostachys rubromarginata* (Rubro), and *Phyllostachys aureosulcata* (Sulcata), were selected and collected bi-weekly over 20 months from established cultivars at the Memphis Zoo Bamboo Farm (Agricenter, Memphis, TN, USA). Fresh live leaves (n = 10) were randomly selected from different rows throughout the plot for each species and NIR spectra (n = 10/ scans per leaf) were taken in the field from both sides of each leaf to create the Fresh–Whole mode dataset (Figure 1A) using an ASD FieldSpec®3 Vis-NIRS portable spectrometer (350–2500 nm, 50 scans with a 34 ms integration time, approximately 1 nm resolution) and a 20 mm in diameter ASD low-intensity 'plant probe' specially designed for heat-sensitive targets such as plant tissues. The selected leaves were collected and dried in an air-forced oven at 60 °C to constant weight for 48 h and pressed for NIR spectra (n = 10 scans/per leaf) to generate the Dry–Whole leaf mode dataset (Figure 1B). Next, dried leaves were coarsely ground with a Wiley mill (Figure 1C) and then finely ground into 0.1 mm isotropic particles with a cyclone mill (Figure 1D, Udy Corporation, Fort Collins, CO, USA). The NIR spectra of dried and ground samples were then collected (n = 3 replicates/ sample). In the standard procedures of this study, each ground sample mode was scanned three times as replicates, as those samples were homogeneous and in small volumes. However, extra scans were collected from different parts of the whole leaves of Dry–Whole and Fresh–Whole sample modes (n = 10 replicates/sample) to cover the structural variation of the tissue in the whole bamboo leaf. All the samples and spectra were classified into four processing modes: the Fine–Ground samples are considered the Master mode, while the minimally processed Coarse–Ground, Dry–Whole, and Fresh–Whole are considered satellite modes. The Fresh–Whole samples are unprocessed, while Dry–Whole and Coarse–Ground samples can be simply processed by a portable dryer and coffee grinder, which are practical in situ (Figure 2). Cyclone milling to obtain the Fine–Ground samples is not feasible for in-field applications as the equipment is not portable, and the samples are easily contaminated or lost during processing.

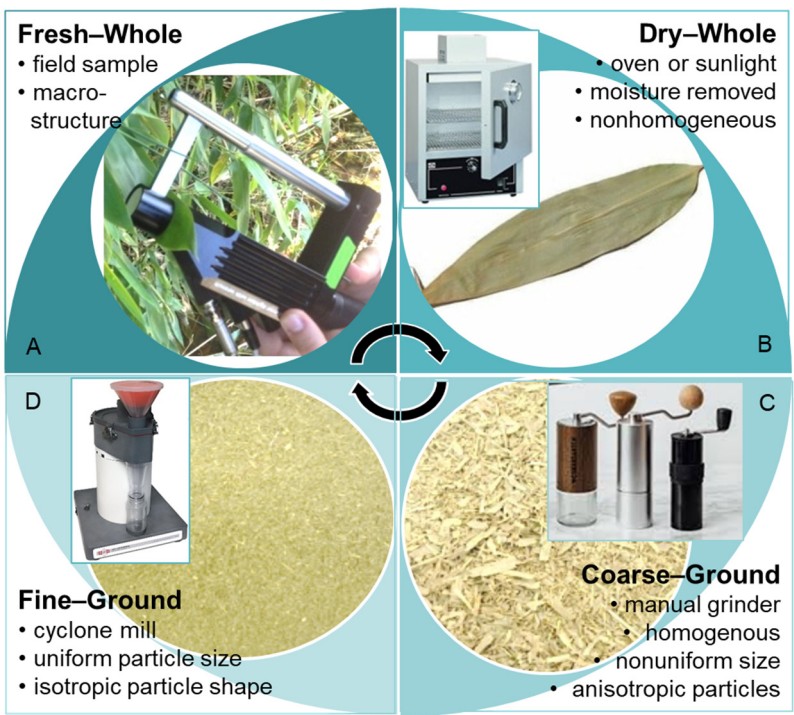

**Figure 1.** Four processing modes of forage samples: (**A**) Fresh–Whole, (**B**) Dry–Whole, (**C**) Coarse–Ground, and (**D**) Fine–Ground.

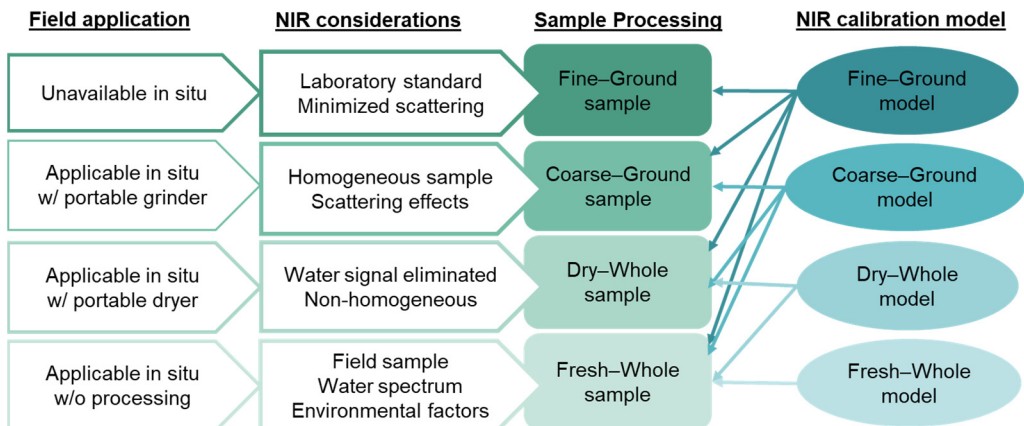

**Figure 2.** Flow chart of the sample processing for NIR data collection from field samples to the most processed for obtaining the best NIR spectra. The sample processing steps eliminate complicating factors in the NIR spectrum, such as overlapping water signals, mixture inhomogeneity, and scattering from variable particle size and shape.

## 2.2. Chemical Analysis

The ground samples were sent for proximate analysis to generate the chemical reference data needed to calibrate the NIR Master mode (Forage Labs, Cumberland, MD, USA). The dry matter (DM) content was determined after the ground subsamples were weighed and dried at 100 °C for 24 h. Crude protein (CP) is the total protein and was determined using the micro-Kjeldahl method [38]. The percentage of nitrogen was calculated using a Leco Nitrogen Analyzer (model FP-2000, Leco Corporation, St. Joseph, MI, USA) (AOAC, 1995), and the percentage of CP was obtained by multiplying the percentage of nitrogen by a standardized factor of 6.25 as protein averages approximately 16 % nitrogen. In addition, a randomized set of samples (n = 10) were split and measured separately to calculate the standard error of the laboratory values (SEL) [39].

*2.3. Chemometric Analysis*

2.3.1. Calibration Models

Before the application of chemometrics, an outlier detection analysis was performed, in which outlier samples were removed from the database. The spectra collected from each leaf under the conditions of each sample processing step were averaged and arranged into four balanced databases (Table 1). The databases were equally weighted by random sampling within each species for calibration and internal validation. Spectra in Database 1 contained equal numbers of samples from the three bamboo species and were randomly divided into calibration (23 samples for each bamboo species) and internal validation sets (6 samples for each species) comprising 80% and 20% of the original samples, respectively. In databases 2–4, spectra from two bamboo species were combined such that 80% of samples were randomly selected to establish calibration models (n = 23 for Rubro and Sulcata, n = 31 for Rubro and Bissetii, n = 23 for Bissetii and Sulcata), and the remaining 20% of samples were used as internal validation sets (n = 47 for Bissetii, n = 29 for Sulcata, n = 39 for Rubro). Spectra collected from the third species were used as the external validation set, with 47 samples from Bissetii, 29 samples from Sulcata, and 39 samples from Rubro in Databases 2–4, respectively. The mathematical pre-treatments of Standard Normal Variate (SNV) with de-trending and a second derivative (Savitzky–Golay smoothing, symmetric points = 16, smoothing points = 33) were applied on the full data spectrum (350–2500 nm). Principal Component Analysis (PCA) and wavelength-range evaluation were applied to the databases to determine dominant peaks in the loadings using the Unscrambler® X v.11, Aspen Technology Inc, MA, USA. The Partial Least Squares Regression (PLSR) algorithm for each processing mode was applied on the transformed spectra associated with the crude protein information (1000–2500 nm) from the balanced databases to predict the crude protein values. The models were calibrated and validated using the calibration set and internal validation set, respectively, while the prediction results were obtained from the external validation set.

**Table 1.** Balanced databases containing NIR spectra collected from each bamboo species under each processing mode. Databases 2–4 are generated using spectra from two bamboo species and the calibration models were tested on spectra collected from the third bamboo species.

| Database | Species | Calibration for Each Processing Mode | Validation for Each Processing Mode | External Validation for Each Processing Mode |
|----------|---------|--------------------------------------|-------------------------------------|----------------------------------------------|
| 1 | Rubro | 23 | 6 | 0 |
| | Bissetii | 23 | 6 | 0 |
| | Sulcata | 23 | 6 | 0 |
| 2 | Rubro | 23 | 6 | 0 |
| | Sulcata | 23 | 6 | 0 |
| | Bissetii | 0 | 0 | 47 |
| 3 | Rubro | 31 | 8 | 0 |
| | Bissetii | 31 | 8 | 0 |
| | Sulcata | 0 | 0 | 29 |
| 4 | Bissetii | 23 | 6 | 0 |
| | Sulcata | 23 | 6 | 0 |
| | Rubro | 0 | 0 | 39 |

2.3.2. NIR Mode Cloning (NIR-MC) Using Slope and Bias Correction (SBC)

The NIR-MC was carried out in Unscrambler® X v.11 (Aspen Technology Inc, MA, USA). The best calibration model for the master (Fine–Ground) mode to predict the crude protein values was applied to the three satellite sampling modes. Additionally, models built with samples from the Coarse–Ground mode were applied to the satellite modes Dry–Whole and Fresh–Whole, and the best-fit calibration model for the Dry–Whole mode was applied to the Fresh–Whole samples (Figure 2). The SBC was performed on predicted

Y-values only, without correcting the spectral matrix. The correction of the predicted values and model evaluation was performed using R studio (Version 1.2.5019, RStudio, Inc., Boston, MA, USA). With the SBC, it is assumed that a linear relationship exists between the predictions of spectra measured from the master (Fine–Ground) mode samples and the predictions obtained from the satellite (Coarse–Ground) samples using the same PLSR model built on the Fine–Ground mode. The SBC was conducted using the external validation set in Database 2 containing the predicted CP values of Fine–Ground and Coarse–Ground samples using the same Fine–Ground PLSR model. Database 2 was divided into two subsets, a training set and a test set, comprised of 70% and 30% of the dataset, respectively. With ordinary least squares, the training set was used to build a linear regression between the predictions of the Fine–Ground samples and the predictions of the Coarse–Ground samples using the same Fine–Ground model:

$$y_{fg} = bias + slope * y_{cg} \tag{1}$$

where $y_{fg}$ is the predicted CP values of Fine–Ground samples using the Fine–Ground calibration model, and $y_{cg}$ is the predicted CP values of Coarse–Ground samples using the Fine–Ground calibration model. Slope and bias coefficients were computed using a linear regression between those two sets of predictions.

Then, in the test set, the predicted values of Coarse–Ground samples were corrected by the slope and bias (intercept) based on the linear equation:

$$y_{cg.\ corr} = bias + slope * y_{cg} \tag{2}$$

where $y_{cg.\ corr}$ is the corrected predictions of Coarse–Ground samples.

The same procedures were applied to the mode cloning from the Fine-Ground model to Dry–Whole and Fresh–Whole samples, from the Coarse–Ground model to Dry–Whole and Fresh–Whole samples, and from the Dry–Whole model to Fresh–Whole samples (Figure 2). The root mean square error of prediction (RMSEP), standard error of prediction (SEP), bias, the coefficient of determination ($R^2$), and the residual prediction deviation (RPD) were used to summarize the accuracy of the prediction. The RPD is the standard deviation (SD) of the sample's reference values divided by the root mean square error of prediction [40]. The RPD value is used as the criteria for evaluating model performance, a model with an RPD value higher than 1.8 is considered a good model for quantitative prediction of a constituent such as crude protein, and RPD > 2.5 indicates an excellent model [41].

### 3. Results

*3.1. Sample Statistics*

The CP content of all the bamboo samples varied from 11.6% to 22.8%, with an average of 17.1% and an SD of 2.1%, determined by the chemical analyses. The frequency distribution plots of CP values are shown in Figure 3. The histogram shows a normal distribution of CP values overlapping across the three bamboo species. The histogram also shows a broad range of CP values was available to build reliable calibration models for predicting CP values in bamboo samples. The Tukey's test ($p > 0.05$) showed no significant difference between species in the means. The standard error of laboratory (SEL) was 0.49, calculated using the technical duplicates of the same samples.

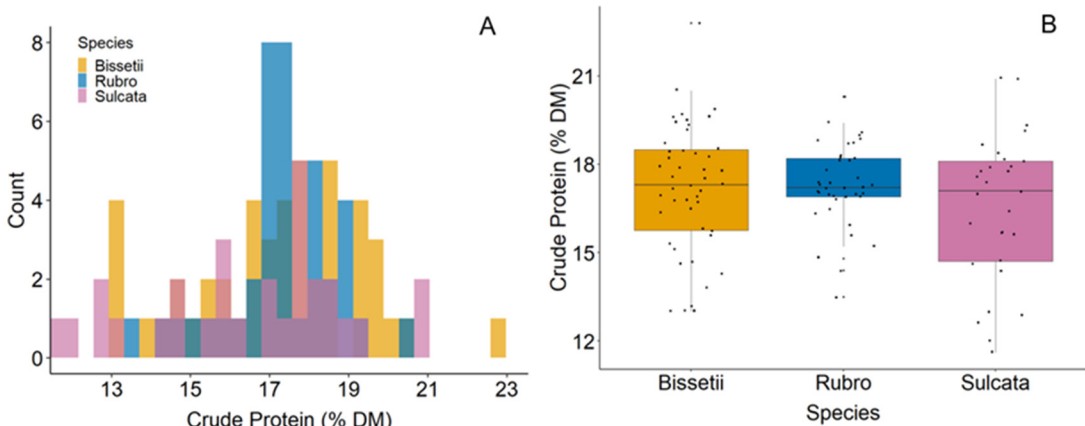

**Figure 3.** Distribution of crude protein of bamboo samples grouped by species presented by the histogram (**A**) and boxplot (**B**).

### 3.2. Spectral Characteristics

Figure 4 shows the averaged raw and transformed spectra collected from the four evaluated processing modes, grouped by species. Several major bands can be observed in the raw spectra. For Fresh–Whole samples, the two prominent peaks appeared near the wavelength ranges of 1460–1500 nm and 1850–1950 nm are the dominant and broad near-IR absorption bands of water [42,43]. The major absorption bands of protein can be observed near the wavelength ranges of 1640–1680 nm, 2050–2100 nm, and 2290–2390 nm, which are assigned to the amide A-amide II combination and high-frequency aliphatic CH stretching bands [44,45].

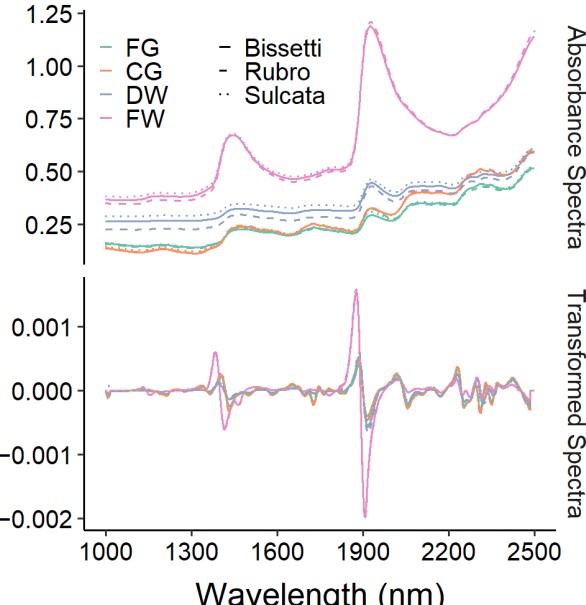

**Figure 4.** Raw (**up**) and transformed (**down**) spectra from four processing modes, grouped by species. Spectra collected by each processing mode from each species were averaged. Fresh–Whole (FW), Dry–Whole (DW), Coarse–Ground (CG), Fine–Ground (FG).

The transformation helped accentuate the spectral response to chemical changes in the samples. Transformed spectra show that the absorbance at around 1425 (first overtone of O-H and N-H in amino and amide groups), 1650–1680 (CH), 1900 (NH), 2050–2060 (ROH/NH), and 2060 (carbonyl stretch of the primary amide) nm have more substantial peaks as the protein content increased [45,46]. These protein signals would be an important factor in building the calibration models for determining CP content in bamboo samples

but are masked by the water signal in Fresh-Whole samples. Thus, the overlapping water signals would have to be factored out in transferring calibration models from dry samples to fresh samples.

The raw and transformed spectra of the three dried modes (Fine–Ground, Coarse–Ground, and Dry–Whole) shared similar major absorption bands. Considering that there is no significant difference between bamboo species in their mean reference CP values suggests that it is possible that SBC without spectra correction between processing modes can provide acceptable results in mode cloning.

### 3.3. NIR Modeling across Species

The PCA scores plots and loadings for all the processing modes are shown in Figures 5 and 6. In the Fine–Ground (Figure 5A), the first two PCs account for 73% of the variance when the three species were not well distinguished, indicating different bamboo species exhibit overlapping confidence intervals. In this case, the PCA loadings (Figure 5B) showing the prominent peaks that explained the trends in the scores plot accounted for PC-1 = 59%, PC-2 = 14% of the variance Similarly, in the Coarse–Ground plot (Figure 5C), the three species of bamboo overlap with PCA loadings explaining 58% and 14% of the variation for PC-1 and PC-2, respectively (Figure 5D). In these two ground modes, the PCA loadings plots (Figure 5B–D) show that wavelengths from the regions 1400–1425, 1900–1950, and 2310–2340 nm are essential for the first and second PC loading values, which are the dominant peaks influencing the trends in the prediction of crude protein. Likewise, in the two whole modes, the scores from the three bamboo species overlap (Figure 6A–C), and the regions 1900–1950 and 2310–2340 nm in the loadings are shown as crucial for PC-1 and PC-2 values (Figure 6B–D).

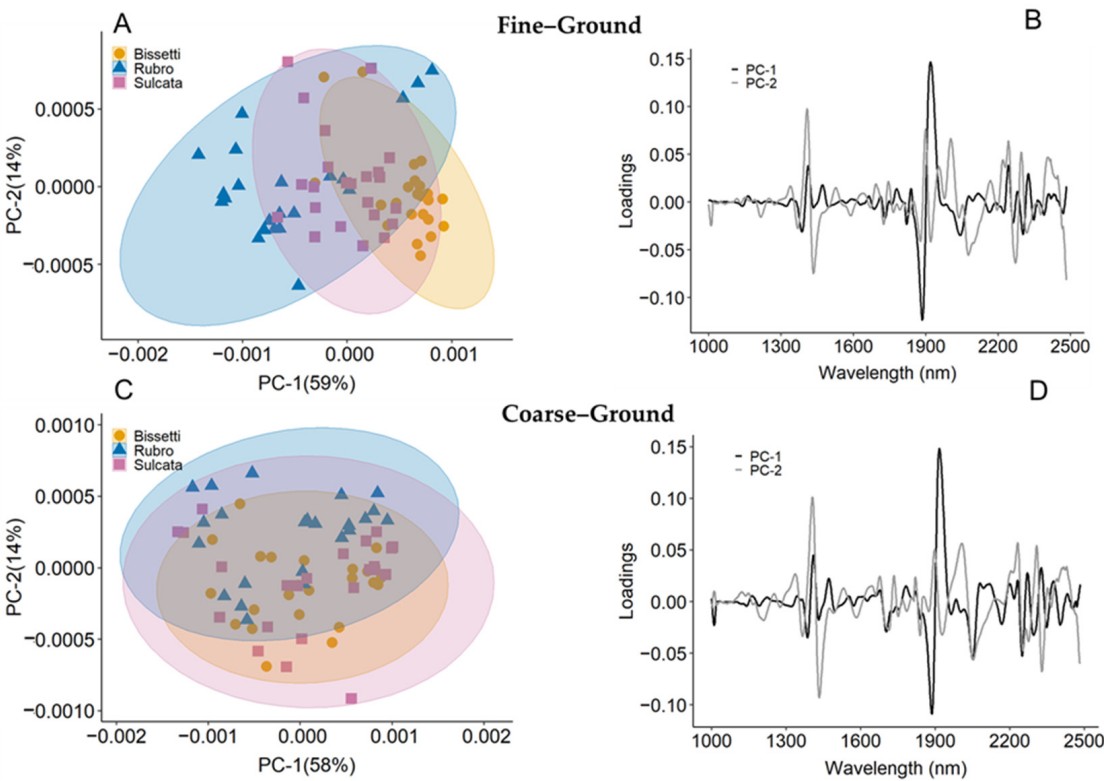

**Figure 5.** PCA scores plot (**A**) and loadings plot (**B**) for spectra collected from Fine–Ground bamboo samples. PCA scores plot (**C**) and loadings plot (**D**) for spectra collected from Coarse–Ground bamboo samples.

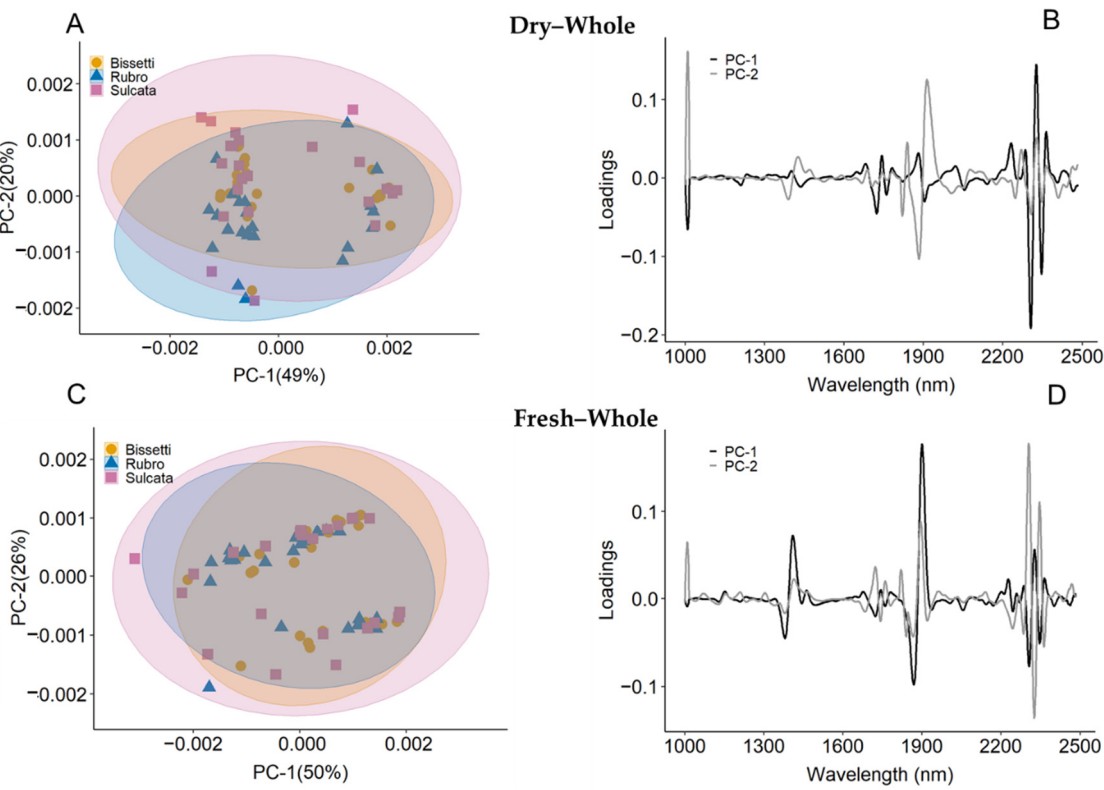

**Figure 6.** PCA scores plot (**A**) and loadings plot (**B**) for spectra collected from Dry–Whole bamboo samples. PCA scores plot (**C**) and loadings plot (**D**) for spectra collected from Fresh–Whole bamboo samples.

The statistical summary of the calibration models for predicting CP across species is shown in Table 2, and contains RMSEC, RMSECV, RMSEP, SEC, SECV, SEP, $R^2$, and RPD values. The main goal was to test if the calibration models based on samples from two species could be directly transferred to the third species of the same genus. For the Fine-Ground, all three calibration models (Databases 2–4) transferred to the third species well, with $R^2$ ranging from 0.82 to 0.99, and RPD ranging from 2.14 to 12.8 in the calibration, cross-validation, validation, and external validation sets, indicating an excellent quantitative predictive performance. The Dry–Whole and Fresh–Whole in Database 4 provided a very poor prediction of CP values, with RPD values lower than 1.0. When Database 4 was excluded, it showed that Coarse–Ground models had good performance in quantitative prediction of CP contents ($0.76 < R^2 < 0.98$, $1.86 < RPD < 9.10$) in the calibration, cross-validation, validation, and external validation sets. Though the Dry–Whole and Fresh–Whole models still performed well in the calibration and cross-validation set ($0.71 < R^2 < 0.92$, $1.78 < RPD < 3.53$; $0.83 < R^2 < 0.97$, $2.50 < RPD < 6.19$, respectively), they failed to provide good performance in the validation and external validation set. The Dry–Whole models had fair performance in predicting CP values ($0.25 < R^2 < 0.75$, $1.04 < RPD < 2.00$) in the validation and external validation sets, whereas the Fresh–Whole models had less accuracy ($0.21 < R^2 < 0.39$, $1.09 < RPD < 1.13$). The ratio of RMSEP and RMSECV (RMSEP/RMSECV) in databases 1–3 ranged from 0.46 to 1.60, 0.78 to 3.13, 0.94 to 2.13, and 1.38 to 3.06 in Fine–Ground, Coarse–Ground, Dry–Whole, and Fresh–Whole mode, respectively. An RMSEP/RMSECV ratio larger than 1.2 might suggest that the acquired calibration set is not large enough to properly capture the variance of the population [47].

**Table 2.** Prediction results of the PLSR models for predicting the quantity of crude protein across bamboo species. The calibration model of each processing mode was applied to its corresponding sample type in all databases. In Database 1, spectra from all bamboo species were combined to generate calibration models. Calibration models for crude protein were built from Sulcata and Rubro and used to predict Bissetii in Database 2; models were built from Bissetii and Rubro and used to predict crude protein in Sulcata in Database 3, and models built from Bissetii and Sulcata were used to predict crude protein in Rubro in Database 4.

| Predicted Species | Calibration Model | PCs | Calibration | | | | Cross-Validation | | | | Validation | | | | External-Validation | | | |
|---|---|---|---|---|---|---|---|---|---|---|---|---|---|---|---|---|---|---|
| | | | R² | RMSEC | SEC | RPD | R² | RMSECV | SECV | RPD | R² | RMSEP | SEP | RPD | R² | RMSEP | SEP | RPD |
| Database 1 —3 species combined | FG model | 6 | 0.97 | 0.34 | 0.34 | 6.29 | 0.96 | 0.43 | 0.43 | 5.00 | 0.88 | 0.69 | 0.70 | 2.89 | | | | |
| | CG model | 9 | 0.97 | 0.38 | 0.38 | 5.70 | 0.94 | 0.57 | 0.57 | 3.79 | 0.82 | 0.88 | 0.89 | 2.26 | | | | |
| | DW model | 9 | 0.90 | 0.63 | 0.64 | 3.25 | 0.80 | 0.94 | 0.94 | 2.20 | 0.42 | 1.48 | 1.52 | 1.34 | | | | |
| | FW model | 9 | 0.96 | 0.36 | 0.36 | 5.28 | 0.83 | 0.75 | 0.76 | 2.50 | 0.22 | 1.78 | 1.84 | 1.11 | | | | |
| Database 2 —Bissetti | FG model | 7 | 0.98 | 0.25 | 0.25 | 8.15 | 0.96 | 0.40 | 0.41 | 5.09 | 0.92 | 0.59 | 0.61 | 3.60 | 0.91 | 0.65 | 0.65 | 3.37 |
| | CG model | 9 | 0.98 | 0.21 | 0.21 | 9.10 | 0.96 | 0.47 | 0.47 | 3.98 | 0.81 | 0.90 | 0.93 | 2.36 | 0.91 | 0.82 | 0.70 | 2.65 |
| | DW model | 6 | 0.85 | 0.77 | 0.78 | 2.58 | 0.71 | 1.12 | 1.13 | 1.78 | 0.75 | 1.06 | 1.11 | 2.00 | 0.68 | 1.26 | 1.16 | 1.73 |
| | FW model | 9 | 0.96 | 0.34 | 0.34 | 5.03 | 0.89 | 0.65 | 0.66 | 2.61 | 0.21 | 1.96 | 2.05 | 1.09 | 0.39 | 1.93 | 1.82 | 1.13 |
| Database 3 —Sulcata | FG model | 7 | 0.96 | 0.29 | 0.29 | 4.92 | 0.93 | 0.39 | 0.39 | 3.64 | 0.93 | 0.47 | 0.46 | 3.58 | 0.96 | 0.70 | 0.70 | 3.59 |
| | CG mode | 9 | 0.97 | 0.27 | 0.27 | 5.86 | 0.92 | 0.45 | 0.46 | 3.50 | 0.95 | 0.36 | 0.38 | 4.59 | 0.76 | 1.34 | 1.37 | 1.86 |
| | DW model | 7 | 0.92 | 0.42 | 0.42 | 3.53 | 0.72 | 0.79 | 0.79 | 1.87 | 0.25 | 1.61 | 1.65 | 1.04 | 0.61 | 1.53 | 1.56 | 1.63 |
| | FW model | 9 | 0.97 | 0.21 | 0.21 | 6.19 | 0.88 | 0.47 | 0.48 | 2.78 | 0.37 | 1.47 | 1.40 | 1.13 | 0.29 | 2.30 | 2.33 | 1.09 |
| Database 4 —Rubro | FG model | 6 | 0.97 | 0.40 | 0.41 | 5.75 | 0.93 | 0.62 | 0.63 | 3.70 | 0.99 | 0.29 | 0.26 | 12.80 | 0.82 | 0.65 | 0.65 | 2.14 |
| | CG model | 6 | 0.97 | 0.41 | 0.41 | 5.86 | 0.94 | 0.59 | 0.60 | 4.06 | 0.76 | 1.88 | 1.91 | 1.95 | 0.84 | 1.20 | 0.56 | 1.16 |
| | DW model | 6 | 0.91 | 0.57 | 0.58 | 3.33 | 0.61 | 1.26 | 1.29 | 1.51 | 0.44 | 2.75 | 2.83 | 1.33 | 0.30 | 1.43 | 1.45 | 0.97 |
| | FW model | 11 | 0.96 | 0.46 | 0.46 | 5.05 | 0.58 | 1.44 | 1.46 | 1.61 | 0.80 | 2.01 | 2.10 | 1.82 | 0.34 | 1.44 | 1.38 | 0.97 |

The root mean square error of calibration (RMSEC), standard error of calibration (SEC), the coefficient of determination (R²), and the residual prediction deviation (RPD). The root mean square error of cross-validation (RMSECV), standard error of cross-validation (SECV), the coefficient of determination (R²), and the residual prediction deviation (RPD). The root mean square error of prediction (RMSEP), standard error of prediction (SEP), the coefficient of determination (R²), and the residual prediction deviation (RPD). Fresh–Whole (FW), Dry–Whole (DW), Coarse–Ground (CG), Fine–Ground (FG).

Thus, it can be concluded that, for the three bamboo species used in this study, the cross-species model transfer can be applied directly without performing any standardization procedure in Databases 2 and 3. However, when transferring the calibration model built for Bissetii and Sulcata samples to Rubro samples in Database 4, only the Fine–Ground and Coarse–Ground modes performed well enough for quantitative prediction of CP, while the two whole-leaf modes all had poor prediction results in the validation set. For further testing of NIR-MC across processing modes, Database 4 was excluded and since Database 2 had higher RPD values in calibration, cross-validation, and validation sets in general, compared with Database 3, it was selected to apply NIR-MC in the following steps.

### 3.4. NIR-MC and SBC across Processing Modes

The prediction statistics of CP values after applying NIR-MC and SBC on Database 2 are shown in Table 3 and Figures 7 and 8. Figure 7 shows that all the transferred models after SBC had lower RMSEP values and increased RPD values than models without SBC. The improved model performances after SBC indicated the necessity of standardizing predicted values in mode cloning. Table 3 shows that, when applying the Fine–Ground Master model to Coarse–Ground bamboo samples, the prediction results were excellent before (RMSEP = 0.83, RPD = 2.62) and after SBC (RMSEP = 0.79, RPD = 2.73). The RPD values increased by 0.74 and 1.07 after SBC when applying the Fine–Ground model to Dry–Whole and Fresh–Whole samples. When transferring the Coarse–Ground model to Dry–Whole and Fresh–Whole samples, RPD values increased after SBC. When applying the Dry–Whole model to Fresh-Whole samples, the prediction results also improved, with the RPD value increasing after SBC.

**Table 3.** Prediction results of the PLSR models for crude protein transferred across processing modes before and after the slope and bias correction (SBC) were applied to Database 2. The calibration model from the Master mode (Fine–Ground) was applied to each of the three satellite modes. The calibration model from the Coarse–Ground mode was applied to the Dry–Whole and Fresh–Whole modes, and the calibration from the Dry–Whole mode was applied to the Fresh–Whole mode.

| Calibration Model | Slope | Offset | Correlation | $R^2$ | RMSEP | SEP | Bias | RPD |
|---|---|---|---|---|---|---|---|---|
| FG sample–FG model | 0.85 | 2.59 | 0.96 | 0.91 | 0.65 | 0.65 | 0.02 | 3.37 |
| CG sample−FG model—before SBC | 0.96 | 0.51 | 0.93 | 0.87 | 0.83 | 0.82 | −0.23 | 2.62 |
| CG sample–FG model—after SBC | 0.82 | 3.19 | 0.93 | 0.87 | 0.79 | 0.80 | 0.16 | 2.73 |
| CG sample–CG model | 0.80 | 3.93 | 0.95 | 0.91 | 0.82 | 0.70 | 0.44 | 2.65 |
| DW sample–FG model—before SBC | 0.60 | 8.43 | 0.85 | 0.73 | 1.99 | 1.19 | 1.62 | 1.10 |
| DW sample–FG model—after SBC | 0.58 | 7.26 | 0.85 | 0.73 | 1.19 | 1.20 | 0.20 | 1.84 |
| DW sample–CG model—before SBC | 0.41 | 12.51 | 0.72 | 0.51 | 2.67 | 1.53 | 2.21 | 0.80 |
| DW sample–CG model—after SBC | 0.31 | 12.40 | 0.72 | 0.51 | 1.62 | 1.62 | 0.38 | 1.32 |
| DW sample–DW model | 0.57 | 7.27 | 0.83 | 0.68 | 1.26 | 1.26 | −0.16 | 1.73 |
| FW sample–FG model—before SBC | 0.70 | −9.30 | 0.66 | 0.44 | 14.42 | 1.77 | −14.32 | 0.15 |
| FW sample–FG model—after SBC | 0.26 | 12.96 | 0.66 | 0.44 | 1.72 | 1.67 | 0.56 | 1.22 |
| FW sample–CG model—before SBC | 0.68 | −13.33 | 0.53 | 0.28 | 19.05 | 2.45 | −18.91 | 0.11 |
| FW sample–CG model—after SBC | 0.19 | 14.41 | 0.53 | 0.28 | 1.85 | 1.86 | 0.36 | 1.16 |
| FW sample–DW model—before SBC | 0.50 | −6.15 | 0.77 | 0.59 | 14.59 | 1.42 | −14.52 | 0.15 |
| FW sample–DW model—after SBC | 0.40 | 10.22 | 0.77 | 0.59 | 1.46 | 1.50 | 0.09 | 1.49 |
| FW sample–FW model | 0.57 | 6.68 | 0.62 | 0.39 | 1.93 | 1.82 | −0.68 | 1.13 |

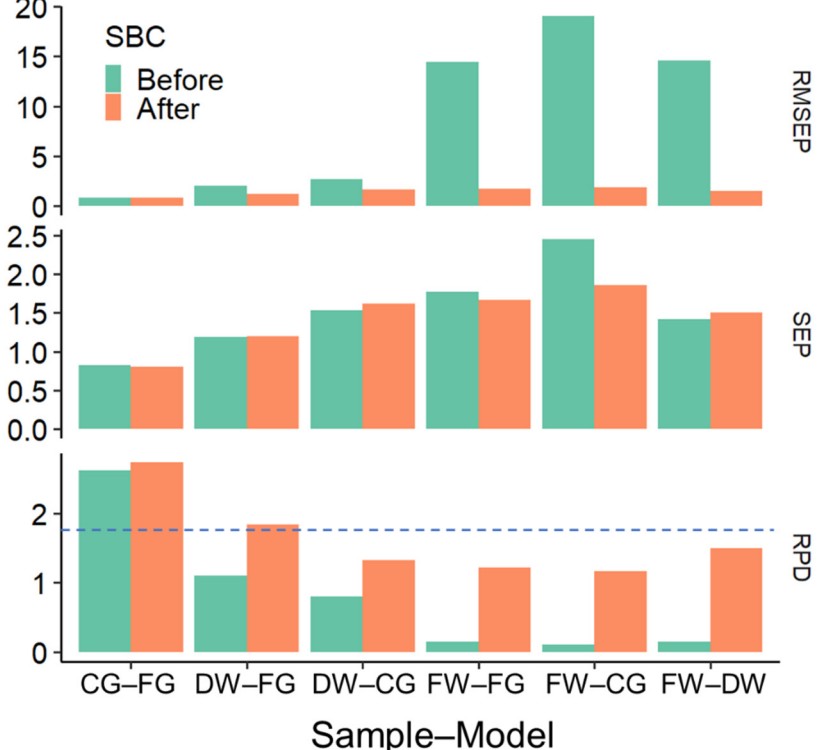

**Figure 7.** The root mean square error of prediction (RMSEP), standard error of prediction (SEP), and the residual prediction deviation (RPD) values before and after slope and bias correction (SBC), predicting samples in satellite modes using Fine–Ground (FG), Coarse–Ground (CG), and Dry–Whole (DW) models. RMSEP and SEP values decrease after SBC, and RPD values increase after SBC.

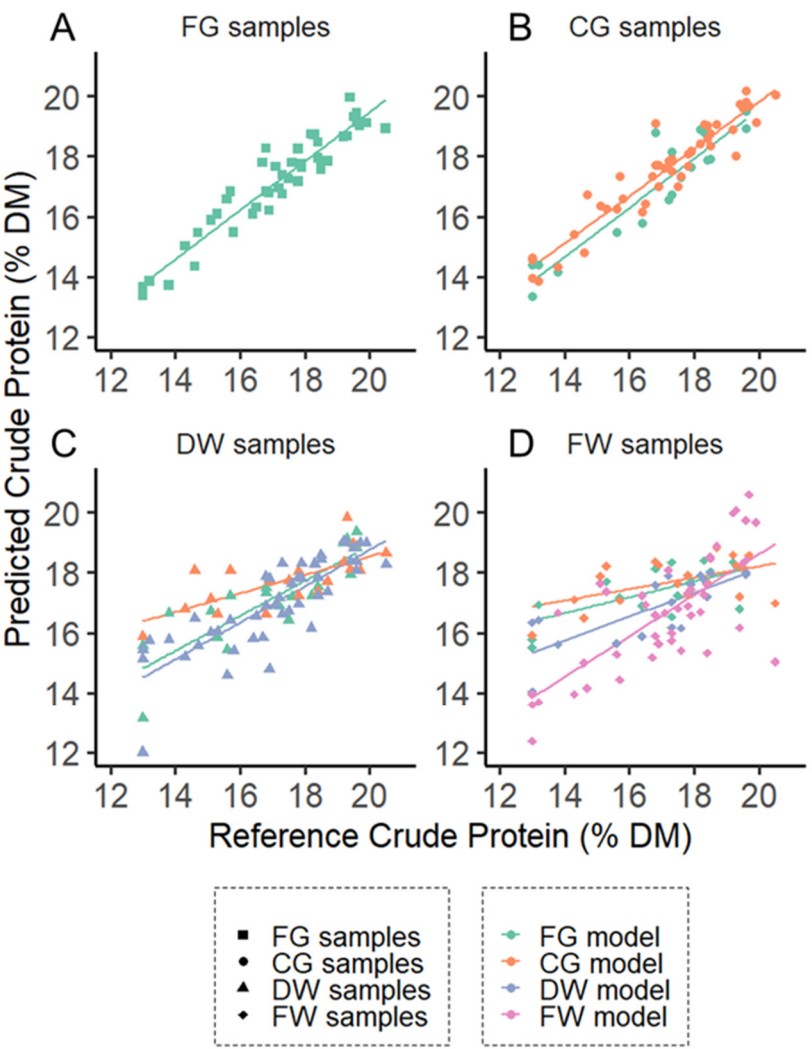

**Figure 8.** Predicted vs. Reference values of CP. Fine–Ground samples using Fine–Ground model (**A**), Coarse–Ground samples using Fine–Ground and Coarse–Ground models (**B**), Dry–Whole samples using Fine–Ground and Coarse–Ground models (**C**), and Fresh–Whole samples using Fine–Ground, Coarse–Ground, and Dry–Whole models (**D**). SBC was used in each mode cloning.

By comparing the performances of the original model generated for each processing mode and the models using the NIR-MC approach (Figure 8 and Table 3), we found the best-fit model for each sampling process. For Coarse–Ground samples, the transferred Fine–Ground model after SBC (RPD = 2.73) had a better performance than the original Coarse–Ground model (RPD = 2.65). For Dry–Whole samples, the transferred Fine–Ground model after SBC (RPD = 1.84) provided better prediction than either the original Dry–Whole model or the transferred Coarse–Ground model after SBC (RPD = 1.73, 1.32). For Fresh–Whole samples, the transferred Dry–Whole model after SBC (RPD = 1.49) was better than the original Fresh–Whole model, the transferred Coarse–Ground model after SBC, and the transferred Fine–Ground model after SBC (RPD = 1.13, 1.16, 1.22). Thus, the transferred Fine–Ground models after SBC were the best-fit models to predict CP values of samples of Coarse–Ground and Dry–Whole samples, while the transferred Dry–Whole models after SBC had the best performance in Fresh–Whole samples.

## 4. Discussion

The NIR-MC approach was used across bamboo species and sample processing modes to determine the CP content in bamboo leaves. The model transfer, and Slope and Bias

Correction (SBC), were applicable to a model generated from Fine–Ground samples to predict CP values of less-processed bamboo samples across species. Constructing a master calibration model based on an extensive database containing different bamboo species and Fine–Ground samples under laboratory conditions that are transferable through NIR-MC allows a fast and accurate CP prediction of new bamboo samples under various circumstances and eliminates the need for creating additional models specific to sample processing states. Currently, NIR mode cloning or calibration transfer techniques are primarily applied to address variation in sample physical or chemical composition due to batch differences, instrumental changes, or environment change over time [22]. Only a few investigators have performed mode cloning from ground samples to non-milled samples using spectral correction methods [19]. This study demonstrates that the simple SBC, without complex spectra correction, was capable of achieving good prediction accuracy when transferring the calibration model from Fine–Ground to Coarse–Ground and even Dry–Whole forage samples. The target subject matter of bamboo leaves makes our study unique and applicable to ecophysiology as this is the first time the NIR-Mode cloning method has been used as a rapid tool in assessing the CP contents of the primary food sources of the giant panda and red panda.

We showed that NIR could be applied directly across bamboo species from the same genus. *Phyllostachys* is one of the easiest bamboo genera to identify and can be defined by its taxonomic and morphological characteristics. For example, compared with the tropical Bambusa group, *Phyllostachys* species are less fibrous [48]. In leaf anatomy, they lack fusoid cells, adaxial ribs, expanded leaf margins, and well-developed abaxial papillae and prickles [49]. Due to the similarities in leaf morphology and chemical composition, the spectra patterns between species are similar, showing no significant separation in the PCA. Furthermore, the distribution and means of CP values do not show significant variance. The similarities in spectra and reference data enable us to use calibration models across species, even in less-processed samples. However, when applying the same models to *Pseudosasa japonica,* which is from a different genus, all the transferred models showed poor performance in predicting CP values (RPD $\leq$ 1.54, $R^2 \leq$ 0.80). We found that the mean amount of CP in *Pseudosasa japonica* (15.32% DM) was significantly different ($p < 0.05$) than in the *Phyllostachys* species (16.35, 17.19, 17.31% DM), which would cause an error in the calibration if this species were added into the spectral databases. A micromorphological study has revealed distinct characteristic variations in the leaf surface between the two genera as well. For example, there are no papillae around the stomata in *Phyllostachys* species, while in *Pseudosasa japonica*, the stomata are surrounded by 4–10 elongated papillae [50]. The differences in micromorphological features might lead to spectra variation between the two genera, affecting the model performance for whole leaves. Therefore, we need a different calibration model for *Pseudosasa* or other genera in future studies.

When considering the RPD value as the criterium for evaluating model performance, a model with an RPD value higher than 1.8 indicates a good model for quantitative prediction of a constituent such as crude protein, and RPD > 2.5 indicates an excellent model [41]. We found that, with the NIR-MC method, the master Fine–Ground model is the best-fit model for all three satellite modes. It can also provide more-accurate predictions than using the original satellite models directly. The RPD values were higher than 2.5 when applying the Fine–Ground model to both Fine–Ground and Coarse-Ground samples with NIR-MC after SBC. This demonstrated that the cloned model has a similar prediction ability in predicting CP contents using Coarse–Ground samples as the laboratory-standard Fine–Ground samples. The Fine–Ground model transferred and standardized to Dry–Whole samples had an RPD value larger than 1.8, indicating that this model met the standard for quantitative prediction. None of the models applied to Fresh–Whole samples were good enough for quantitative predictions, with RPD < 1.8 (Figure 7).

In this study, we have established a database containing three bamboo species and four levels of processing. Although the Fine–Ground model applied to Fine–Ground samples yields the best accuracy in predicting protein content, it is time-consuming and requires

a laboratory cyclone mill grinder, which is not amenable to field studies. We also show that the Fine–Ground Master model has good performance in predicting protein content of the satellite modes Coarse–Ground and Dry–Whole. Therefore, the Fine—Ground model obtained from the existing databases built in the laboratory can be applied to new field samples of satellite processing modes by NIR-MC using SBC. According to various environmental and practical situations, we can choose between processing modes to obtain predictive results of CP content. The drying process is required to obtain good predictions of CP values as the water signal overlaps that of protein. However, Dry–Whole bamboo leaves can be acquired by drying with a portable drier or natural sunlight. To obtain higher-quality predictions of CP values with similar prediction ability as the laboratory standard, grinding the dried samples with a portable coffee grinder is necessary. This Coarse–Ground processing can be performed in situ using dried samples.

The differences between the evaluated processing modes affected the performance of the NIR-MC. The calibration model transferred from Fine–Ground to Coarse–Ground mode had the best performance of predicting CP values because Coarse–Ground samples were similar to Fine–Ground samples in particle size and distribution. The differences between Dry–Whole and Fine–Ground modes were more significant as the Dry–Whole samples were not homogenous and still maintained the original physical structure of the bamboo leaves, which can cause more significant variation in spectra dependent on where and how the spectra are collected from the leaves. However, all the dry modes (Fine–Ground, Coarse–Ground, and Dry–Whole) shared a similar spectral pattern and similar major absorption bands. The similarity of spectra patterns presented in the three dried modes indicated that a simple univariate correction of slope and bias is enough without the spectral correction. The moisture in Fresh–Whole samples caused the variation of spectra, particularly in the shifts of some absorption bands related to water, thus limiting the application of NIR-MC by having strongly biased predictions. To remove the effects of moisture from NIR spectra, a mathematical algorithm is needed to project all the spectra of wet samples orthogonally to the space of unwanted variation [51]. We propose that external parameter orthogonalization (EPO) or a model transfer tool of combining direct orthogonal signal correction with slope and bias correction (DOSC–SBC) [52] can be applied on Fresh–Whole samples to remove the spectral variation caused by water before using the NIR-MC.

We chose to analyze the CP of bamboo leaves as a nutritional parameter of the diets of giant and red pandas because it affects their forage behavior and seasonal movement. Both species maintain a carnivorous digestive tract but have evolutionarily adapted to a diet of bamboo; as such, these two foragers depend on the changes in the nutrient content distribution in the bamboo plant based on species and season [32,36]. Dietary proteins that giant and red pandas can obtain from bamboo leaves are limited because the bamboo leaf is a high-fiber and low-nutrition food. To meet this nutritional requirement, during the adaptation to becoming bamboo specialists, specific genes helped by increasing the efficiency of releasing lysine and arginine from dietary proteins and amino acid recycling, thus offsetting the limited protein content in the bamboo diet [53]. Despite the low protein content (7.0% to 17.9%) in bamboo leaves, energy utilization of giant pandas is sourced primarily from protein (61%), with the remainder split between carbohydrates (23%) and fats (16%) [54]. Red pandas also have a foraging behavior to increase protein intake by selecting certain bamboo species with higher protein content [55]. The macronutrient composition of giant and red pandas' diets with high dietary protein-energy is similar to other carnivore diets, showing the importance of assessing the CP values of bamboo as an indicator of their diet quality. The NIR-MC technique provides real-time assessment of protein values without the time-consuming procedures of chemical analysis or sample processing, potentially allowing rapid in-field decisions for the management of panda habitat based on forage structure and nutrient patterns.

Crude protein is not the only necessary nutrient for giant and red pandas. Previous studies found that the changes in carbohydrates (culm starch and bound glucose) of bamboo

caused the changes in giant pandas' foraging behavior during spring [36]. The changes in calcium, phosphorus, and nitrogen concentrations also affected giant panda annual migration patterns as they switch their diet between the leaves and shoots of bamboo species. The NIR-MC is not limited to crude protein analysis, nor is it limited to analysis of bamboo, but has potential for application of other nutrient parameters and other forage systems, such as that of the eucalyptus and koala [56].

## 5. Conclusions

In this study, the viability of the NIR-MC approach was examined in predicting the crude protein values of bamboo leaves. With databases containing different bamboo species and sample processing modes, we demonstrated the possibility of using NIR across different bamboo species sharing similar leaf morphology. We also demonstrated the potential of NIR-MC in transferring the Master mode to less-processed satellite modes. For future studies in the evaluation of bamboo nutrition, with the help of NIR-MC and SBC, new spectra of field samples processed with a portable dryer and grinder could be fitted into the existing database of the master model, allowing a real-time prediction of diet quality available to the giant panda and red panda.

**Author Contributions:** Conceptualization, C.K.V.; methodology, Q.S., X.O. and C.K.V.; formal analysis, Q.S., C.K.V. and M.S.-R.; validation, Q.S.; investigation, Q.S., X.O. and C.K.V.; resources, A.J.K. and C.K.V.; data curation, Q.S., M.S.-R., X.O. and C.K.V.; writing—original draft preparation, Q.S.; writing—review and editing, Q.S., M.S.-R., A.J.K. and C.K.V.; visualization, Q.S. and C.K.V.; supervision, A.J.K. and C.K.V.; project administration, C.K.V.; funding acquisition, A.J.K. and C.K.V. All authors have read and agreed to the published version of the manuscript.

**Funding:** This project was supported by a USDA U.S. Forest Service International Programs cooperative agreement, grant # 18-DG-11132762-248, on 'Coupled Forest-Human Ecosystems: Connecting Forest Management with Conservation Initiatives', the U.S. Department of Agriculture, Agricultural Research Service, Biophotonics project # 6066-31000-015-00D, and the Mississippi Agricultural and Forestry Experiment Station, the National Institute of Food and Agriculture, U.S. Department of Agriculture, Hatch project under accession number W3173.

**Data Availability Statement:** Data are available upon request from Carrie K. Vance.

**Acknowledgments:** We thank the giant panda staff of the Memphis Zoo for providing information about the bamboo species in the giant pandas' diet.

**Conflicts of Interest:** The authors declare no conflict of interest.

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
