# Peer review of "Near-Infrared Spectroscopy and Mode Cloning (NIR-MC) for In-Situ Analysis of Crude Protein in Bamboo"

_remotesensing, doi:10.3390/rs14061302_

Round 1
Reviewer 1 Report
This paper presents the results of an experimentation, consisting in testing the ability of near infrared spectroscopy (NIRS) coupled to chemometrics, to assess crude protein content (CP) in bamboo leaves, in outdoor conditions.
More specifically, the feasibility of : (i) predicting the CP of several bamboo species (ii) predicting the CP of bamboo leaves having undergone different sample preparation, ranging from dried and ground, to raw leaves, outdoor. For this purpose, the authors chose to investigate the calibration transfer of a database on dried and ground products, using the bias/slope correction (SBC).
Although the application to bamboo is new, the general problem has been widely studied, particularly in the field of forages. In addition, a number of methodological errors and wrong messages make this proposed article fail to meet the standards of the journal in my opinion.
In more detail, I noted the following problems:
- line 16, among others: the term "satellite" has been chosen to denote the non-laboratory measurement modes (dried/ground). This term is confusing, especially in the context of the journal "remote sensing". It would be preferable to use the term "target" or "slave", which are accepted terms in the NIRS community.
- line 47-49: the authors state that light scattering affects the slope of NIRS spectra. However, scattering has much more complex effects, including the flattening of the spectra (fine powders appear white, regardless of their colour).
- line 64-65: I don't know where the term "NIR Mode Cloning" comes from. I only found it in an oral communication by the same authors. It seems to be a confidential name. It would be more appropriate to speak of calibration transfer, a term that is widely used and recognised by the NIRS and chemometrics community.
- lines 70-99: too much detail is given, which is not useful in the rest of the article.
- lines 171-174: the piling up of pretreatments seems to be the result of a very empirical approach, without any reflection of a spectroscopic nature. In particular, the baseline offset correction is totally redundant with the second derivative.
- paragraph 2.3.2: why did you choose the SBC correction, which is one of the most inefficient to compensate for the complex effects of scattering? Many other methods exist, such as optical standardisation (PDS), orthogonalisation (EPO), the RepFile method, ...
- line 194: what does "standardisation of the predicted values" mean?
- line 224-226: the statement delivered is completely wrong. Just because the Y distributions match does not mean that the calibration can be transferred. Conditions must also be met for the spectra.
- lines 252-255: Many conclusions are proposed, following the examination of the factor map of the first two components of a PCA. However, looking at the first two components of a PCA can tell us that two populations are separated, but not that they are similar. They may separate on the third component, or the tenth...
- Table 3: there are obviously calculation errors, because we should have RMSEP^2=SEP^2+Bias^2, which is not the case at all.
Reviewer 2 Report
The total number of samples, the number of samples in calibration and validation must be better explained.
The criteria used to split the data set into calibration and validation must be added.
The standard error of the lab method must be added.
More details on the calibration transfer protocol should be added.
Reviewer 3 Report
a standard instrument, a standard commercial program and a standard method are used. The kind of examination is already done for many other stuff and products. The only novelty is the material. perhaps no other study is done with bamboo leaves.
For me the new ideas and sexy approaches are missing.
What is new in data evaluation, or preprocessing or postpr
Reviewer 4 Report
Review report for manuscript: remotesensing-1561795
This manuscript by Sheng et al. (‘Near-Infrared Spectroscopy and Mode Cloning (NIR-MC) for 2 in-situ analysis of crude protein in bamboo’) is a comprehensive and interesting study towards the development of a rapid analytical method based on NIR spectroscopy for assessment of the nutritional quality of bamboo leaves. The explored application is related to a wider context, as protein content determination remains one of the most essential analyses performed in various scenarios. Noteworthy, the accomplishments of this study have more universal reach (i.e. as the topic of calibration transfer remains very active in the field of analytical spectroscopy), but in particular the selected objects of the study make it unique and interesting – as those constitute to the primary diet of giant pandas and red pandas. Thus, the scope of this study finds good scientific and practical justification and meets the interest of this journal.
The manuscript is generally very well arranged and properly written, and provides helpful information for the interested readers.
The selection of the methods and their use seems appropriate, the analysis and interpretation of the results is interesting, and noted should be an exhaustive discussion (presented in Section 4), extending far beyond the routine one often seen in similar papers.
That being said, a few minor points could be explained and/or addressed by the authors, which should bring this work to a level fully adequate to be accepted for publication in Remote Sensing. The extent of changes fit the criteria of a minor revision.
- Analysis of RMSEC/RMSECV ratios presented in Table 2, for some cases could suggest that the acquired calibration set is too small to properly capture the variance of the population (i.e. the models might be prone to overfitting issue). Perhaps the authors would like to address this somehow in their paper.
- It is not entirely clear why different scan averaging numbers (3 and 10) were applied.
- SG smoothing is given without the symmetric part and the central point (it is in fact 33). This should be clear, but perhaps not immediately for all readers (as this is the convention of presenting the stencil size used by Unscrambler, but not necessarily universal). Perhaps this could be presented in a clearer way (again, just to prevent some potential misunderstandings).
- Some more recent literature could help position this work better in the current development trends, e.g. protein analysis by miniaturized NIR spectrometers remains an active field of research (e.g. Molecules 2021, 26, 6390). Given the rapidly growing importance of miniaturized, portable NIR sensors, it would perhaps be worth to point the readers to more up-to-date sources (e.g.: Eur. J. 2021, 27, 1514).
Round 2
Reviewer 1 Report
This paper presents the results of experimentation, consisting in testing the ability of near-infrared spectroscopy (NIRS) coupled to chemometrics, to assess crude protein content (CP) in bamboo leaves, in outdoor conditions.
In this new version, the authors only partially answer my questions and criticisms.
They continue to show a certain naivety towards fundamental concepts of chemometrics (Q6 Q8, Q9). They persist in narrow concepts, for example by considering that what they do is not calibration transfer (Q3).
They often hide behind citing a few authors who have made the same mistakes as they have. For example, they continue to assert (Q6) that the Bias and Slope correction is as efficient as the other calibration transfer methods, relying on the citation of works that have shown this equivalence in particular cases. However, it is obvious, mathematically speaking, that the SBC is a zero-level correction and that it is inherently less efficient than more sophisticated methods, such as PDS, TOP.
Reviewer 3 Report
the authors did a huge amount and succesfull work to improve the manuscript.